# The Role of Pro-Opiomelanocortin Derivatives in the Development of Type 2 Diabetes-Associated Myocardial Infarction: Possible Links with Prediabetes

**DOI:** 10.3390/biomedicines12020314

**Published:** 2024-01-29

**Authors:** Nompumelelo Anna-Cletta Gumede, Andile Khathi

**Affiliations:** Department of Human Physiology, School of Laboratory Medicine and Medical Sciences, College of Health Sciences, University of KwaZulu-Natal, Durban X54001, South Africa; khathia@ukzn.ac.za

**Keywords:** pro-opiomelanocortin, catecholamines, glucocorticoids, opioids, type 2 diabetes, prediabetes, myocardial infarction

## Abstract

Myocardial infarction is a major contributor to CVD-related mortality. T2DM is a risk factor for MI. Stress activates the HPA axis, SNS, and endogenous OPS. These POMC derivatives increase the blood glucose and cardiovascular response by inhibiting the PI3K/AkT insulin signaling pathway and increasing cardiac contraction. Opioids regulate the effect of the HPA axis and SNS and they are cardioprotective. The chronic activation of the stress response may lead to insulin resistance, cardiac dysfunction, and MI. Stress and T2DM, therefore, increase the risk of MI. T2DM is preceded by prediabetes. Studies have shown that prediabetes is associated with an increased risk of MI because of inflammation, hyperlipidemia, endothelial dysfunction, and hypertension. The HPA axis is reported to be dysregulated in prediabetes. However, the SNS and the OPS have not been explored during prediabetes. The effect of prediabetes on POMC derivatives has yet to be fully explored and understood. The impact of stress and prediabetes on the cardiovascular response needs to be investigated. This study sought to review the potential impact of prediabetes on the POMC derivatives and pathways that could lead to MI.

## 1. Introduction

Myocardial infarction (MI) is the sudden death of myocardial tissue due to hypoxia and ischemia following a blockage of a coronary vessel. The coronary vessel is often blocked by an atherosclerotic or thrombotic plug growth or rupture [1]. Ischemia starves the cardiomyocytes of nutrients and causes metabolic and iconic imbalances in the cells and, ultimately, death [2]. Despite the numerous interventions to reduce the mortality of CVD, it is still the most significant contributor to global mortality [3]. In 2019, 32.9% of the global 15.6 million deaths were due to CVD [4]. According to the Global Disease Burden, in 2019, ischemic heart diseases (IHDs) accounted for 197.2 million prevalent cases and 9.1 million deaths [5]. The contributing risk factors to IHD include hypertension, low-density lipoprotein cholesterol (LDL-C), diabetes mellitus, and smoking [6]. These risk factors are also prevalent in type 2 diabetes mellitus (T2DM) [7]. Psychosocial stress as a risk factor for CVD is often overlooked, yet in observational studies, chronic work stress is associated with a 40–60% increase in the occurrence of coronary heart diseases [8]. In patients with MI and unstable angina, a history of economic stress and social isolation significantly increased the risk of cardiovascular events [9]. The hyperactivity of the hypothalamus–pituitary–adrenal axis (HPA axis), sympathetic nervous system (SNS), and increased inflammation are some of the mechanisms that are postulated to link psychosocial stress and the development of MI [10]. Β-endorphin is simultaneously released with adrenocorticotropic hormone (ACTH) from pro-opiomelanocortin (POMC) when the HPA axis response is activated by stress and is involved in attenuating acute stress responses [11]. Chronic stress increases the risk of developing T2DM, CVD, and mental health disorders [12]. The HPA axis and the SNS are hyperactivated in T2DM and have been linked with the development of CVD [13,14]. Studies have reported that T2DM-related disorders begin during prediabetes, which is a condition that precedes T2DM [15]. Prediabetes is an asymptomatic state in which the blood glucose levels are above normal but below the threshold of T2DM diagnosis [16]. Prediabetic individuals with severe perceived stress are reported to have high HbA1c levels [17]. This indicates that stressed prediabetic individuals are at a high risk of progressing to T2DM and its complications [18]. Prediabetes is reported to be an independent risk factor for MI and increases the likelihood of experiencing MI by 25% [19]. The review aims to discuss the potential impact of prediabetes on POMC derivatives and pathways that could lead to MI.

## 2. POMC Derivative Signaling

### 2.1. Glucocorticoids

Glucocorticoids (GCs), such as cortisol in humans and corticosterone (CORT) in rodents, are cholesterol-derived hormones secreted by the zona fasciculata of the adrenal glands [20]. Glucocorticoids are fundamental in maintaining resting and stress-related homeostasis [21]. Animals respond to stress by activating various behavioral and physiological responses collectively called the stress response [22]. As summarized in Figure 1, the principal effectors of the stress response are localized in the paraventricular nucleus (PVN) of the hypothalamus, the anterior lobe of the pituitary gland, and the adrenal gland. This structure collection is commonly called the hypothalamic–pituitary–adrenocortical (HPA) axis. In response to stress, corticotropin-releasing hormone (CRH) is released into hypophysial portal vessels that access the anterior pituitary gland. The binding of CRH to its receptor on pituitary corticotropes induces the release of adrenocorticotropic hormone (ACTH) into the systemic circulation. The adrenal cortex is the principal target for circulating ACTH, which stimulates glucocorticoid synthesis and secretion from the zona fasciculata [22]. The cellular availability of GCs is determined by two enzymes that have opposing effects: 11β-hydroxysteroid dehydrogenase type 2 (11βHSD2) oxidizes cortisol into the inactive metabolite cortisone, whereas 11β-hydroxysteroid dehydrogenase type 1 (11βHSD1) converts cortisone to cortisol. Therefore, in normal physiology, GC levels are tightly regulated by a negative feedback loop at the level of the hypothalamus and pituitary gland, the availability of CBG in circulation, and at target tissues through the action of 11βHSD1 and 11βHSD2 [23].

Both endogenous and pharmacological GCs act through a dual system formed of corticosteroid receptors: the GC receptor (GR/NR3C1) and the mineralocorticoid receptor (MR/NR3C2) [25]. Upon activation through the binding of GC, GR undergoes a conformational change and dissociates from accessory proteins such as heat shock proteins (HSP-90, p60/Hop, HSP-70) and migrates to the nucleus, where the GC/GR complex binds to a promoter region called the glucocorticoid response element (GRE), causing an increase or decrease in transcription [26,27]. GCs also have a non-genomic action in which they do not bind to GR but rather to transmembrane receptors and activate mitogen-activated protein kinases (MAPKs), adenylyl cyclase (AC), protein kinase C (PKC), and heterotrimeric guanosine triphosphate-binding proteins (G proteins) [27,28]. The redox state during oxidative stress drives cardiomyocyte and vascular smooth muscle cell MR activation by GCs [29]. The functions of GC can be summed up as follows: Firstly, it prepares the metabolic, autonomic, psychological, hemostatic, and cardiovascular components of the stress response. This involves other stress hormones such as catecholamines, glucagon, and angiotensin II (ANGII) through the stimulation of alpha and beta-adrenergic receptors and the ANGII receptor. Secondly, it prevents inflammation, cellular proliferation, and the tissue repair process from over-activation, leading to injury. Thirdly, it prepares the body for prolonged nutrient deprivation by facilitating muscle proteolysis [30]. Though GCs are essential for maintaining homeostasis, excessive or impaired glucocorticoid secretion can adversely affect the body [31]. Increased GC levels due to chronic stress, exogenous therapy, or endocrine disorders commonly lead to atherosclerosis hypertension and increase the risk of developing cardiomyopathies [32].

### 2.2. Catecholamine

Catecholamines are a group of molecules that act as neurotransmitters and hormones in the sympathetic division of the autonomic nervous system [33]. Catecholamines and glucocorticoids are the principal hormones secreted in response to extrinsic or intrinsic stressors to maintain homeostasis [34]. Catecholamines are produced from tyrosine hydroxylation to DOPA (l-3,4-dihydroxyphenylalanine) and a series of cellular reactions that ultimately produce dopamine (D), norepinephrine (NE), and epinephrine (E) from the adrenal medulla [35]. The SNS and adrenal medulla release epinephrine. It is involved in several physiological functions, including regulating blood pressure, vasoconstriction, cardiac stimulation, and blood glucose levels [36]. Norepinephrine is mainly produced by neurons within the locus coeruleus (LC) and takes part in diverse motor and mental functions, including locomotion control, motivation, attention, cognition, and memory formation [37]. Catecholamines rapidly respond to stress by binding to adrenergic receptors at the threatened site and the alerted brain, heart, and muscles [34,35,38]. Adrenoceptors (ARs) are categorized into alpha (α1 and α2) and beta (β1, β2 and β3) receptors. Norepinephrine activates α-AR and β1-AR, while epinephrine activates all subtypes of α- and β-AR [39]. The α1-AR receptors bind to stimulatory Gq proteins, activate phospholipase C (PLC), and induce constriction. On the contrary, α2-AR receptors are coupled to G inhibitory (Gi) proteins that inactivate adenylyl cyclase (AC), decreasing cyclic adenosine monophosphate (cAMP) production [40]. Β1-AR predominates in the heart and binds to the G stimulatory (Gs) protein–AC–cAMP–protein kinase A (PKA) signaling cascade, which results in the phosphorylation of troponin I, the L-type Ca^2+^ channel, phospholamban (PLN), and the cardiac ryanodine receptor (RyR), thus resulting in increased cardiac contraction and relaxation [41]. Β2-AR is distributed extensively throughout the body but is expressed predominantly in bronchial smooth muscle cells [39]. 

The stimulation of β2-AR activates the Gi protein, which inhibits cAMP and activates mitogen-activated protein kinase (MAPK) and cytosolic phospholipase A2 (cPLA2), thus resulting in cAMP-independent Ca^2+^ enhancement and reduced cardiac contraction [41]. β3-ARs are abundantly expressed in white and brown adipose tissue, increasing fat oxidation, energy expenditure, and insulin-mediated glucose uptake [39]. Β3-AR is stimulated by catecholamines only at high doses and has negative inotropy (decreased contraction) by facilitating the nitric oxide synthase (NOS) pathway. Nebivolol is reported to restore hemodynamic properties in patients with heart failure by stimulating β3-AR [42]. Figure 2 illustrates catecholamine adrenergic signaling.

### 2.3. Opioids

The endogenous opioid peptides consist of endorphins, dynorphins (DYNs), and enkephalins (ENKs) [43,44]. There are four types of endorphins: alpha (α), beta (β), gamma (γ), and sigma (σ) endorphins. β-endorphins are primarily synthesized and stored in the anterior pituitary gland from their precursor protein, pro-opiomelanocortin (POMC) [43]. Dynorphin (DYN) is derived from a precursor protein, prodynorphin. Prodynorphin is cleaved to yield dynorphin-B, which has two extended forms (dynorphin-A and dynorphin-B), and leumorphin. In the peripheral circulation, dynorphin-A and dynorphin-B are further cleaved to yield dynorphin A (1–13) (DYN-A(1–13)), dynorphin A (1–8) (DYN-A (1–8)), and dynorphin B (1–13) (DYN-B (1–13)) and rimorphin, respectively [45]. Enkephalins are derived from the precursor protein proenkephalin (PENK) and interact with glutamate and dopamine in the brain reward circuit [46]. Endogenous opioids activate the mu (µ) (MOR), kappa (κ) (KOR), and delta (δ) (DOR) opioid receptors [43,44]. Opioid receptors signal through G-protein coupled receptors by stimulating inhibitory G-proteins, thus causing the Gαi subunit to dissociate from the Gβγ subunit and inhibit cAMP production. The Gαi subunit also interacts with the G-protein gated inwardly rectifying potassium channel (Kir3), thus causing hyperpolarization. The Gβγ modulates Ca^2+^ conduction by reducing the activation of N-type, P/Q-type, and L-type Ca^2+^ channels [47]. Signaling through the Gα_i/o_ coupled proteins causes negative inotropy in rats’ ventricular myocytes by inhibiting cAMP-dependent Ca^2+^ [44]. The activation of KOR activates Gα_i/o_ proteins, inhibiting the AC production of cAMP and releasing Gβγ, which modulates the conduction of Ca^2+^ and K^+^ channels [48]. Dynorphin provides cellular protection through the Gαs/cAMP/PKA signaling pathway, which causes an increase in CREB phosphorylation to enhance cell proliferation [49]. The distribution of OPR throughout the limbic system is consistent with the role of endogenous opioids in attenuating stress [50]. The stress system activates the dopaminergic reward system and the amygdala, thus forming a positive feedback loop [51]. The CRH stimulates the hypothalamus to release an α-melanocyte-stimulating hormone (α-MSH) and β-endorphin from the POMC-containing neurons in the arcuate nucleus. The α-MSH and β-endorphin inhibit CRH and the LC/NE system, thus regulating the stress response [51].

The naturally occurring δ-opioid peptide agonist, leucine enkephalin (LE), is co-released with the β-AR agonist norepinephrine (NE) from the nerve terminals in the heart during sympathetic stimulation. LE inhibits NE-induced increases in sarcolemma L-type Ca^2+^ current, cytosolic Ca^2+^ transient, and contraction [52]. The “anti-stress” activity of endogenous opioids may be mediated explicitly by the MOR [50]. During acute stress, the MOR regulation of the LC opposes the excitatory effect of CRH and protects against the detrimental NE/E hyperactivity effects. These opposing functions promote recovery after stress termination [53]. Mice with selective deletion of β-endorphin, enkephalin, or dynorphin subjected to the zero-maze test show increased anxiety-related behavioral responses [54]. Corticosterone plasma levels rapidly increased in all strains and returned to baseline after 60 minutes in b-endorphin-deficient mice [54]. 

In contrast, mice lacking dynorphin and enkephalin showed longer-lasting elevated corticosterone levels, which delayed the stress reaction termination [54]. Overall, upon release, opioids oppose the effect of glucocorticoids and catecholamines by inhibiting Ca^2+^ and Na^+^ channels while activating K^+^ channels. Figure 3 illustrates cardiac opioid signaling.

## 3. Role of POMC Derivatives in Cardiovascular Function

### 3.1. Glucocorticoids

Glucocorticoids are essential for the embryonic development of the heart and the maintenance of normal myocardial function. Adrenalectomized rats with GC insufficiency have a reduction in contractile force generation by the heart papillary muscle [55]. Dexamethasone (DEX), an exogenous GC, increases contractility tension and accelerates contraction velocity and relaxation in cardiac muscle [56]. The hearts of GR-null and smooth muscle-specific knockout (KO) mice exhibit irregularly shaped and disorganized myofibrils at the embryonic stage. Furthermore, the expression of genes that are critical for cardiac development and function is diminished in the heart [57]. GC protects from atherosclerosis and inflammation [58]. Mice lacking endothelial GR develop severe atherosclerotic lesions in the aorta and have heightened inflammation in the lesions [59]. 

High-dose corticosteroids are reported to exert cardiovascular protection through the non-genomic activation of eNOS. The binding of CORT to the GR stimulates PI3K and Akt, leading to eNOS activation and NO-dependent vasorelaxation. Furthermore, acute administration of pharmacological CORT concentrations in mice leads to decreased vascular inflammation and reduced myocardial infarct size following ischemia and reperfusion injury [60]. Overall, GCs are needed for the structural development of the heart and protect the heart from inflammation and atherosclerosis.

### 3.2. Catecholamines

The activation of the SNS results in the release of catecholamines, which increase the supply of energy and oxygen delivery [61]. In fight-or-flight mode, NE/E stimulates glycogenolysis, gluconeogenesis, and aerobic glycolysis, inhibiting glycogen synthesis to supply glucose to vital organs [62]. Epinephrine increases blood glucose by stimulating glucagon secretion from mouse α-pancreatic cells through the activation of the α1 and β-AR on the α-pancreatic cell [63]. Glucagon, in turn, stimulates ACTH-induced cortisol release [64]. The stimulation of β-AR in adipocytes activates lipolysis through the activation of adenylyl cyclase and a cascade of reactions, which leads to the phosphorylation of hormone-sensitive lipase (HPL) and adipose triglyceride lipase (ATGL) [65]. It has been reported that catecholamine-induced lipolysis inhibits glucose uptake by inhibiting the PI3K–Akt–mTOR pathway [66]. Plasma concentrations of cortisol and E are significantly elevated in infants with severe hypoglycemia [67]. These mechanisms provide an acute increase in glucose levels.

The spread of electrical impulses from cardiac autorhythmic cells stimulates the myocardium to contract, thus enabling the heart to pump blood to the blood vessels [68]. The electrical impulses are initiated by the sinoatrial (SA) node and result in atrial depolarization and atrial contraction; the impulse is then conducted to the internodal pathway, the AV node, the AV bundle, the left and right branches of the bundle of His, and lastly, the Purkinje fibers, which result in ventricular depolarization and contraction [68]. The activation of β1-AR increases the SA node’s firing rate, which, in turn, increases contractility because of an increased Ca^2+^ release from the sarcoplasmic reticulum and conduction through the AV node. The stimulation of α1-AR stimulates vasoconstriction and increases peripheral resistance [69]. The net effect is increased BP and cardiac output (CO) due to enhanced cardiac excitation, impulse conduction, and cardiac contraction [70]. Overall, catecholamines increase blood glucose by inhibiting glucose uptake while stimulating glucose release and synthesis. Catecholamines also increase blood pressure by increasing the firing rate of the SA node, Ca^2+^ concentration, and vasoconstriction. 

### 3.3. Opioids

The cardiovascular regulatory effects of endogenous opioids were initially considered to originate from the central nervous system. However, opioid peptides of myocardial origin have been shown to play essential roles in the local regulation of the heart [71]. A portion of POMC mRNA that contains a sequence for β-endorphin is expressed in the cardiac muscle, thus indicating that β-endorphin is produced in the heart [72]. Cell culture experiments from neonatal rat hearts revealed that myocytes and non-myocytes express ppENK mRNA [73]. Β-endorphin (1–31) is the primary endorphin form present in the cardiac muscle, although substantial amounts of N-acetylated and des-acetyl p-endorphin-(1–27) and P-endorphin-(1–26) are also detectable [74]. The cardiac tissue of rats subjected to immobilization stress has elevated β-endorphin [75]. Cardiac MOR expression is elevated in chronic heart failure and plays a cardioprotective role by reducing infarct size, the phosphorylation of ERK, and glycogen synthase-3-β (GSK3β) [76]. Infusions of high-dose β-endorphin in hypertensive subjects are reported to cause a decrease in systemic vascular resistance, blood pressure, plasma NE, and endothelin-1(ET-1) and raise atrial natriuretic factor P (ANP), thus protecting the heart [77]. The stimulation of OPR not only inhibits cardiac excitation–contraction coupling but protects the heart against hypoxic and ischemic injury [71]. Myocardial methionine-enkephalin levels increase with the severity of hypoxic stress in congenital cardiac disease. They may play an essential adaptive role in countering adrenergic over-activity and related excess demand on myocardial metabolic capacity [78]. Dynorphin provides cardiac protection during hypertension. Intracerebroventricular injections of β-endorphin and dynorphin A (1–13) in anesthetized rats result in hypotension and bradycardia [79]. Dynorphin A (1–13) also modulates epinephrine-induced cardiac arrhythmias by increasing the threshold for or suppressing the manifestation of the induced cardiac arrhythmias [80]. Receptor-dependent and independent stimulation of the adrenergic signaling pathway is reported to cause an increase in ppENK mRNA and modulate the dromotropic response to catecholamine stimulation in rat myocardial cells [73]. In stress-induced cardiac injury, the activation of central MOR with endogenous opioids is reported to aggravate stress-induced cardiomyopathy, while the stimulation of peripheral µ-opioid receptors produces a cardioprotective effect [81]. Overall, opioids counteract the effects of catecholamines on the heart. Opioid receptors are elevated during hypoxia and cardiac ischemic injury and attenuate the extent of cardiac damage. Figure 4 summarizes the cardiovascular function of POMC derivates.

## 4. Myocardial Infarction

Myocardial infarction (MI) is cell death due to prolonged oxygen and nutrient deficiency [1]. Myocardial cell death could be via apoptosis, necrosis, or autophagy. Apoptosis is characterized by cytoplasm shrinking, membrane blebbing, nuclear chromatin condensation, chromosomal DNA fragmentation, and the formation of apoptotic bodies. Necrosis, conversely, is characterized by the enlargement of organelles and the rupture of cellular membranes, releasing cytoplasmic and nuclear content and causing an inflammatory reaction [82]. An experimental study investigating the mechanisms of MI and ischemia–reperfusion injury found that ischemic necrosis and inflammation are the primary mechanisms of myocardial death in MI. In contrast, apoptosis is the primary mechanism in ischemia–reperfusion injury [83]. In human heart samples, increased expression of caspase 3 and the presence of apoptotic bodies proved apoptosis to be the pathway of cardiomyocyte death post-MI [84]. Post-MI, MR activation in the brain contributes to sympathetic hyperactivity and increased cardiac aldosterone. Furthermore, MR activation enhances apoptosis and inflammation in myocytes and non-myocytes in the peri-infarct and infarct areas, which central MR blockade reduces [85].

Necrotic myocardial cells are reported to activate high-mobility group box-1 (HMGB1) protein, a ubiquitous death-associated molecular protein (DAMP), which in turn stimulates myocardial inflammation and fibrosis in vivo through the Akt and MAPK pathway [86]. Myocardial injury upregulated the mRNA levels of HMGB3, HIF1α, p65, Reg1α, Reg3γ, and COL18a1 on the third and seventh day in the left anterior descending coronary artery ligation model of myocardial ischemia compared to sham animals. This study discovered that HMGB3 plays a dual role during the progression of myocardial ischemia and infarction, as it plays a pro-inflammatory role and improves cardiac function during the cardiac remodeling phase [87]. The NLRP3/caspase 1/IL-1β pathway is also activated in MI and results in inflammation and fibrosis. The suppression of the pathway by phloretin inhibits inflammation and fibrosis and reverses structural remodeling [88]. Early inflammation is beneficial for cardiac function as it removes necrotic material and heals the infarct area. However, excessive inflammation is detrimental as it contributes to ventricular arrhythmias and heart failure, thus resulting in high mortality in MI patients [89]. These studies highlight apoptosis, necrosis, inflammation, and fibrosis as the critical pathways in MI.

### Risk Factors of Myocardial Infarction

MI is often preceded by atherosclerosis and thrombosis [90]. Atherosclerosis is a build-up of a fat-rich plaque that contains oxidized low-density lipoprotein (LDL) and macrophages [91]. Atherosclerosis is initiated by a chronic increase in reactive oxygen species (ROS) and other inflammatory molecules, which cause vascular damage [92]. Factors that contribute to acute myocardial infarction (AMI) include non-modifiable risk factors such as age, sex, race, and family history. Modifiable risk factors include dyslipidemia, hypertension, smoking, and diabetes mellitus [93]. In AMI patients, Peak-cTnI (a protein released upon myocardial damage) and elevated HbA1c and angiotensin-converting enzyme (ACE) are reported to be significant predictors of AMI. These markers are also positively associated with malondialdehyde (MDA), a marker of oxidative stress and rate pressure products (heart rate × systolic blood pressure), but negatively associated with antioxidant enzyme glutathione peroxidase (GPx), thus indicating the role of oxidative stress in AMI [94].

It has been reported that psychological stress may lead to myocardial ischemia, MI, arrhythmias, and cardiac death [10]. In individuals with stable coronary heart disease, mental stress-induced ischemia is strongly associated with an increased risk of cardiovascular death or nonfatal myocardial infarction [95]. Interestingly, the higher perceived stress level in ACS patients was not associated with a worse prognosis [96]. Chronic POMC stress response system activation has deleterious cardiac effects [97]. Chronic activation of the stress response system may lead to insulin resistance and T2DM, and the presence of stress in T2DM aggravates T2DM and T2DM-related disorders. The following paragraphs will delve into T2DM-associated MI.

## 5. T2DM-Associated MI

### 5.1. Role of Glucocorticoids in T2DM-Associated MI

The activity of the HPA axis is elevated in T2DM [98]. Individuals with T2DM are reported to have elevated basal plasma cortisol, higher levels after DEX suppression, and a more significant response to CRH, indicating a hyperactive HPA axis and impaired negative feedback [99]. Urine and serum CORT, as well as ACTH, hypothalamic Orexin-A, OX2R, CRH, and pituitary ACTH, are reported to be heightened in HFD- and STZ-induced T2DM Sprague Dawley rats compared to non-diabetic rats. This study highlighted the elevation of the HPA axis hormones in T2DM [100]. Elevated plasma cortisol levels are associated with raised fasting blood glucose and total cholesterol and the prevalence of IHD, independently of conventional risk factors [101]. Participants with CAD are reported to have elevated hair cortisol (HCC) compared to those without CAD. Higher HCC was associated with diabetes, hypertension, and hyperlipidemia, which, in turn, were associated with CAD. The standard modifiable risk factors, diabetes, hyperlipidemia, and hypertension mediate the association between ln(HCC) and CAD [102]. The heightened cortisol response to mental stress is associated with myocardial injury, as it is associated with detectable plasma levels of cTnT in healthy participants independently of coronary atherosclerosis [103]. Interestingly, genetically predicted cortisol was found to be unrelated to IHD, ischemic stroke, T2DM, CVD risk factors, or vice versa [104]. These studies report that, in T2DM and CAD, cortisol is increased. The increased cortisol is associated with cardiac injury, as evidenced by elevated cTnT levels. 

### 5.2. Role of Catecholamines in T2DM-Associated MI

In T2DM, there is a reduction in the parasympathetic response and the hyperactivity of sympathetic tone [105]. Hyperglycemia, as observed in T2DM, has deleterious effects on the ANS. In a study by Tarvainen et al., hyperglycemia was associated with a moderate increase in mean HR and a decrease in heart rate variability. In contrast, the duration of diabetes was strongly associated with a reduction in HRV. The most significant decline in HRV related to diabetes occurred within the first 5–10 years of the disease [106]. The hyperactivity of the SNS is also reported in patients with essential hypertension and T2DM or T2DM only, and this constitutes the mechanism of an increased CVD risk in patients with EHT and T2DM [107]. Sustained activation of the SNS in obese and obese and T2DM individuals leads to arterial blood pressure elevation, which triggers arterial damage and cardiovascular events. Furthermore, it contributes to the deterioration of renal diseases [108]. High insulin levels, as in T2DM, act upon the hypothalamic region of the brain, resulting in increased sympathetic outflow and renin release via the β-1-AR stimulation of the renal juxta-glomerular apparatus. High renin levels result in ANG II production and increased sympathetic flow [109]. Hyperglycemia increases the level of dopamine in the striatum and hippocampus, as well as the elevation of norepinephrine in the hippocampus. It increases the epinephrine level in the hypothalamus, midbrain, and pons medulla [110]. There is also an increase in urinary norepinephrine excretion in hyperglycemic and or hyperinsulinemic obese subjects compared to normal subjects. This explains the effect of insulin on the activation of the SNS [111].

#### 5.2.1. Calcium Overload

Catecholamines’ β-AR-induced PKA activation increases Ca^2+^ concentration via the phosphorylation of L-type Ca^2+^ channels, RyR2, and PLN [112]. Furthermore, the stimulation of β-AR activates Ca^2+^/calmodulin-dependent protein kinase II (CaMKII), which phosphorylates Ca^2+^ channels and consequently results in Ca^2+^ overload [112,113]. Calcium overload causes arrhythmias and decreases the force of contraction [114]. It is reported that SR Ca^2+^ leak via RyR2, but not the type 2 inositol 1,4,5trisphosphate receptor (IP3R2), results in mitochondrial Ca^2+^ overload in murine post-MI heart failure models. Mitochondrial Ca^2+^ overload causes ROS generation that oxidizes RyR2 and promotes SR Ca^2+^ leak in failing hearts, resulting in a vicious cycle [115]. Catecholamine-induced mitochondrial Ca^2+^ overload causes oxidative stress in three ways: (1) the oxidative deamination of catecholamines; (2) the activation of nicotinamide adenine dinucleotide phosphate (NADPH) oxidase, which in turn produces superoxide anion radicals in heart muscle cells; and (3) the auto-oxidation of catecholamines, forming aminochromes [116]. Calcium overload results from the increased phosphorylation of calcium channels and receptors. Consequences of calcium overload include arrhythmia and oxidative stress. Oxidative stress may lead to atherosclerosis and MI.

#### 5.2.2. Cardiotoxicity

In experimental rats, injections of isoproterenol (ISO), a non-selective β-AR agonist, induced cardiac toxicity, evidenced by ST elevation; decreases in PR, QRS, and RR intervals; and a significant elevation of biomarkers of cardiac injury, such as aspartate aminotransferase (AST), alanine transaminase (ALT), creatinine kinase (CK-MB), and lactate dehydrogenase (LDH) [117]. Isoproterenol injection is also reported to result in impaired left ventricular function, myocyte death, significant RyR2 hyperphosphorylation, and RyR2-calstabin dissociation. Administrations of JTV519, a RyR2 stabilizer, prevented the ISO-induced death of adult myocytes in vitro. This indicates that hyperadrenergic stimulation causes myocyte death through Ca^2+^, which is caused by defective RyR2 [118]. The PKA phosphorylation of RyR2 Ser-2808 reduces the binding affinity of the channel-stabilizing subunit calstabin2, resulting in leaky RyR2 channels [119]. Using RyR2-S2808A mice, it is reported that PKA phosphorylation of Ser-2808 in the RyR2 channel results in cardiac dysfunction after MI [120]. These experimental studies show that hyperadrenergic stimulation causes defects in RyR2, ventricular dysfunction, and myocyte death.

### 5.3. Role of Opioids in T2DM-Associated MI

The effect of T2DM on endogenous opioids and the role of endogenous opioids in T2DM are not entirely understood. It is reported that liraglutide’s glucose-lowering effects are mediated by the MOR receptor, as the elevation of MOR also elevates PI3K, GSK3β, and GLUT4 expression, and the results are diminished in the presence of naloxone. This study showed that BER is downregulated in hyperglycemia, and its elevation has glucose-lowering effects [121]. Increased adrenal gland secretion of β-endorphin may have a role in ameliorating hyperglycemia by increasing the expression of muscle glucose transporters and decreasing hepatic gluconeogenesis [122]. In insulin-resistant rats, loperamide treatment improves insulin resistance in an MOR-dependent way. Additionally, in MOR knockout mice, insulin resistance induction with a high-fructose diet is more rapid [123]. On the contrary, a study comparing opium-addicted and opium-non-addicted T2DM individuals found that opium usage had no statistically significant effect on blood glucose, HbA1c, FBG, LDL, and diastolic blood pressure. However, systolic BP and the prevalence of high SBP were significantly higher in the opium-user group. According to this study, opium does not seem to have beneficial effects on diabetes control or cardiovascular risk factors [124].

The short-term consumption of opium is reported to increase blood glucose through an increase in hepatic glucose production and a decrease in peripheral glucose extraction, including increases in epinephrine, glucagon, and cortisol. Prolonged consumption of opium is said to result in β-cell failure and insulin resistance; therefore, long-term opioid consumption may aggravate T2DM [125]. A study investigating the effects of methadone dependency on blood glucose, lipids, and glucose-modulating hormones in male and female Wistar rats revealed that only in male rats did methadone significantly lower serum glucose and triglyceride levels; this was linked to a decrease in the amount of hormones that counter-regulate carbohydrate metabolism [126]. These studies show that endogenous opioids and exogenous opioids behave differently in the presence of insulin resistance and hyperglycemia. The studies also reported contradicting results on the effects of opioids on normoglycemia. More research needs to be conducted to investigate the impact of T2DM on endogenous opioids, changes in the endogenous opioids in the presence of T2DM and stress, and how this influences cardiovascular response. Figure 5 summarizes the effects of T2DM on POMC derivates and pathways that could lead to MI.

## 6. Prediabetes

Prediabetes is an asymptomatic state of intermediate hyperglycemia with glucose levels above normal but below the T2DM threshold [127]. Prediabetes is characterized by impaired fasting blood glucose (IFG), impaired glucose tolerance (IGT), and glycated hemoglobin (HbA1c) [128]. The prevalence of prediabetes varies due to the differences in the diagnosis of prediabetes by the ADA, the WHO, and the IDF [129]. Using IGT, the IDF estimated the global prevalence to be 373.9 million, or 7.5%, in 2019. Respectively, 8.0% (453.8 million) and 8.6% (548.4 million) of the population are predicted to have IGT by 2030 and 2045 [130]. Notably, the age range of 20–39 accounts for approximately one-third (28.3%) of persons with IGT, meaning that they will likely spend many years at high risk of developing diabetes and CVD [130]. The consumption of high-calorie diets, a sedentary lifestyle, and being overweight or obese are among the contributing factors to the vast increase in the prevalence of prediabetes [131]. Chronic consumption of an unhealthy diet is associated with increased obesity in animals and humans. This increase in adiposity leads to low-grade inflammation and insulin resistance [132]. Insulin resistance results in impaired glucose homeostasis and plays a role in the pathogenesis of CVD. Impaired glucose homeostasis plays a role in CVD through inflammation, atherosclerosis, endothelial dysfunction, and hypertension [133]. Studies have demonstrated that during prediabetes, there is an increase in heart rate, blood pressure, CRP, and ET-1 and a decrease in eNOS; therefore, there is an increased risk of MI [134]. Impaired fasting glucose (IFG) is associated with unrecognized myocardial infarctions in a multi-ethnic population free of baseline cardiovascular disease [135].

Patients with myocardial infarction with nonobstructive coronary arteries (MINOCA), prediabetes, and diabetes had a significantly higher incidence of major adverse cardiovascular events (MACEs) compared to normoglycemia. In addition, prediabetes had a similar impact to DM on long-term prognosis in patients with MINOCA [136]. Chronic stress results in insulin resistance and T2DM [137]. Stress in the presence of T2DM aggravates T2DM and elevates the risk of developing CVD [14,137]. It is reported that during prediabetes, there is an impaired HPA axis and poor stress response [138]. Opioid peptides and their receptors are upregulated in patients with prediabetes, depending on the significance of IR and the immune cytokines [139]. This study investigated opioids only in prediabetes. The influence of stress and prediabetes on the cardiovascular response is unknown. The SNS is also activated by stress and activates the CVS [140]. Chronic activation of the SNS precedes the development of insulin resistance and prediabetes [141]. The following sections will discuss how the risk factors associated with prediabetes are also associated with MI and how they influence POMC derivatives.

## 7. Prediabetes-Associated MI Risk Factors

### 7.1. Obesity

Obesity is excess lipids in adipose and peripheral tissues due to excess nutrition and a sedentary lifestyle [142]. Chronic high-fat diet (HFD) consumption is associated with increased obesity in animals and humans. This increase in adiposity can impair peripheral insulin sensitivity [132]. Obesity is also associated with chronic low-grade inflammation that causes insulin resistance [143]. The role of obesity in insulin resistance is through the release of free fatty acids (FFAs) and inflammatory mediators. The binding of FFAs to Toll-like receptors (TLRs) downregulates PI3K and Akt, reducing GLUT4 and insulin response [133]. The downregulation of PI3K and Akt results in decreased NO production by eNOS, contributing to hypertension and an increased risk of atherosclerosis [144]. Obesity aggravates myocardial mitochondrial dysfunction following AMI by increasing the mRNA expression of caspase 9/3, Cyt-c, and PARP and decreasing the mRNA expression of PI3K, Bad, and Akt [145]. In a study investigating the cardioprotective properties of roselle in a diet-induced obesity rat model with MI, there was a significant deterioration in cardiac systolic function (LVDP and LVdP/dtmax), diastolic function (LVdP/dtmin and Tau), coronary flow, and cardiac output (RPP) in the obese (OB) and obese with MI (OB + MI) groups compared to the control group. The OB and OB + MI group also showed oxidative stress, as shown by increased *NOX2* gene expression and 8-isoprostane levels, as well as decreased endogenous antioxidant SOD activity and GSH concentration compared to control rats’ hearts [146]. In a retrospective cohort study consisting of men and women with severe obesity and insulin resistance, severe obesity was associated with an increased risk of MI in both women and men without chronic inflammation but not in women and men with inflammation [147]. Subjects with abdominal obesity have a hyper-responsive HPA axis because of a reduced feedback mechanism of ACTH [148]. The altered HPA axis in abdominal obesity is linked with insulin resistance [148]. Cortisol release and the gene expression of 11β-HSD1 are elevated in the adipocytes of obese subjects [149]. In obesity, the adipocytes secrete high levels of adipokines and inflammatory cytokines, which cause neuroinflammation and dysregulation of the HPA axis [150]. Obesity also increases the SNS activity through hyperleptinemia [151]. Leptin activates the POMC melanocortin 3 and 4 receptors (MC3/4R), stimulating the SNS [152]. MOR is reported to be low in obese individuals and increases after weight loss [153]. These studies reveal that obesity results in insulin resistance by releasing FFA, low-grade inflammation, and downregulating the PI3K and Akt pathways. Obesity aggravates myocardial dysfunction by releasing caspase 9/3, Cyt-c, and Bad proteins and increases oxidative stress. The HPA axis and SNS negative feedback are dysfunctional, whereas the opioid MOR is reported to be low in obese individuals.

### 7.2. Hyperlipidemia

Hyperlipidemia is the elevation of lipids and lipoproteins in the blood [154]. The lipoprotein profile includes low-density lipoprotein (LDL) cholesterol, high-density lipoprotein (HDL) cholesterol, and triglycerides [155]. Hyperlipidemia involves the blood’s imbalanced cholesterol levels, including LDL and HDL [156]. LDL-containing ceramides reduce insulin-mediated glucose uptake, Akt phosphorylation, and GLUT4 translocation [157]. HDL removes cholesterol from peripheral tissues for excretion via the liver. The accumulation of cholesterol or impaired cholesterol excretion from the pancreatic β-cell due to a reduced removal of HDL results in a reduced islet number and size, reduced insulin content, and insufficient insulin secretion [158]. An imbalance between HDL and LDL increases the risk of cardiovascular events, including MI and stroke [156]. According to observational studies, LDL positively correlates with MI, while HDL cholesterol negatively correlates with MI [159]. ROS and ox-LDL impair endothelial cells and platelets to increase plasminogen activator inhibitor-1 (PAI-1) expression [160]. PAI-1 mediates the formation of platelet-rich clots that are resistant to fibrinolysis. PAI-1 inhibits both t-PA and u-PA, thereby reducing plasmin generation and favoring thrombin formation, which occludes the vessel [161]. Chronic intake of saturated (but not the monounsaturated) HFD results in elevated plasma CORT and a decreased expression of several GC-related genes in the PVN. HFDs also heighten the HPA responses to DEX [162]. Overall, elevated LDL reduces insulin-mediated glucose uptake. A high level of LDL is associated with an increased risk of MI. An increased expression of PAI-1 due to high LDL could be one reason for the increased MI risk. A few studies have reported on the effect of hyperlipidemia on POMC derivatives.

### 7.3. Inflammation

The normal inflammatory response is acute and involves activating immune and non-immune cells [163]. Pro-inflammatory cytokines can induce insulin resistance and contribute to vascular injury and atherosclerosis [164]. TNF-α, IL-6, and leptin activate inhibitory molecules such as the suppressor of cytokine signaling 3 (SOCS3) and c-Jun NH2-terminal kinase (JNK) that suppress insulin signaling, thus resulting in insulin resistance [142]. IL-6 induces insulin resistance by reducing GLUT4 and IRS1 expression by activating the JAK-STAT (mediates cellular inflammatory response and cellular signals, such as insulin growth factor) and increasing SOCS3 expression [143]. IL-6 stimulates the secretion of C-reactive protein (CRP) by macrophages and T cells [165]. CRP inhibits NO’s release and is strongly associated with an increased risk of atherosclerosis in CVD, independently of cholesterol levels [166]. Proinflammatory cytokines, such as TNF-α, IL-1, and IL-6, stimulate GC release by stimulating CRH, ACTH, and the adrenal cortex. GC suppresses the immune system’s pro-inflammatory cytokine release [167]. Inflammatory cytokines activate NF-kβ by activating the nuclear factor-kβ kinase (IKK) complex inhibitor, which phosphorylates inhibitor I-κB [168]. Activating NF-kβ results in the transcription of many genes, including pro-inflammatory genes [168]. Increased pro-inflammatory cytokines result in the hyperactivation of the HPA axis [169]. Isoproterenol-induced β-AR activation stimulates inflammation in the heart through caspase 1 activation and the activation of IL-18 by the NLRP3 inflammasome [170]. Chronic inflammation causes insulin resistance by activating SOCS3 and JNK, suppressing insulin signaling. GC suppresses the release of pro-inflammatory cytokines by tightly regulating NF-κβ. However, pro-inflammatory cytokines can activate inflammatory cytokine gene transcription via NF-κβ. Β-AR stimulation causes cardiac inflammation.

### 7.4. Endothelial Dysfunction

Endothelial dysfunction is characterized by an imbalance between the endothelium’s vasodilator and vasoconstriction products. Mechanisms that promote vasoconstriction, thrombosis, and inflammation predominate [168]. Endothelium dysfunction is an early sign of atherosclerosis and a CVD risk factor [171]. In prediabetic patients, there was an elevation of urinary 8-hydroxy-2-deoxy-guanosine (8-OHdG), indicating increased DNA damage and homocysteine release from endothelial cells, as well as increased oxidative stress, as characterized by the decrease in the reduced glutathione (GSH)-to-oxidized glutathione (GSSG) ratio (GSH/GSSG). These pathological changes in the endothelium may promote atherogenesis and the development of CVD [172]. Prediabetes and hypertension induce endothelial dysfunction and inflammation by elevating ICAM-1, P-selectin, and TNF-α [173]. In Zucker obese fatty (ZOF) prediabetic rats, TNF-α is reported to induce endothelial dysfunction, as the administration of TNF-α in coronary arteries caused the expression of NAD(P)H, increased superoxide anions, and inhibited endothelium-induced vasodilation via eNOS and abrogated endothelium-dependent dilation [174]. In an HFHC diet-induced prediabetes model, there was a significant increase in ET-1 and a significant decrease in eNOS [175]. ET-1 regulates sympathetic innervation through increased expression and nerve growth factor (NGF) release via the activation of the ET-A receptor on cardiac myocytes [176]. Endogenous ET-1 is reported to have a sympatho-excitatory effect through the ET-A receptor in both normotensive and hypertensive subjects, contributing to basal sympathetic vasomotor tone [177]. The upregulation of ET-1 positively correlates with glutathione reductase activity and ROS but negatively correlates with GPx activity [178]. Overall, prediabetes destroys endothelial cells and results in elevated ET-1 production. ET-1 causes excessive vasoconstriction and hypertension and is associated with oxidative stress. These changes are atherogenic and increase the risk of MI. The literature does not elucidate the influence of elevated ET-1 on POMC derivatives.

## 8. Conclusions

This review outlined that in T2DM, there is a dysregulation of the HPA axis and SNS, thus causing an elevation of GC and catecholamines [179]. Chronic elevation of GC and catecholamine activates pro-atherogenic and pro-inflammatory pathways and causes hypertension, Ca^2+^ overload, oxidative stress, and contractile dysfunction, respectively. These pathways increase the risk of MI. This review also highlighted that prediabetes is associated with obesity, inflammation, hyperlipidemia, endothelial dysfunction, and hypertension, all of which increase the risk of MI. In the literature, the effect of T2DM on endogenous opioids and the impact of prediabetes on glucocorticoids, catecholamines, and opioids are scantly reported. The number of people with prediabetes is on the rise, and so is the number of people experiencing psychosocial stress. Therefore, future studies should investigate (1) the impact of prediabetes on the HPA axis, SNS, and OPS; (2) the concurrent effect of prediabetes and stress on the CVS; and (3) endogenous opioid signaling during stress and its cardiovascular protective role.

## Figures and Tables

**Figure 1 biomedicines-12-00314-f001:**
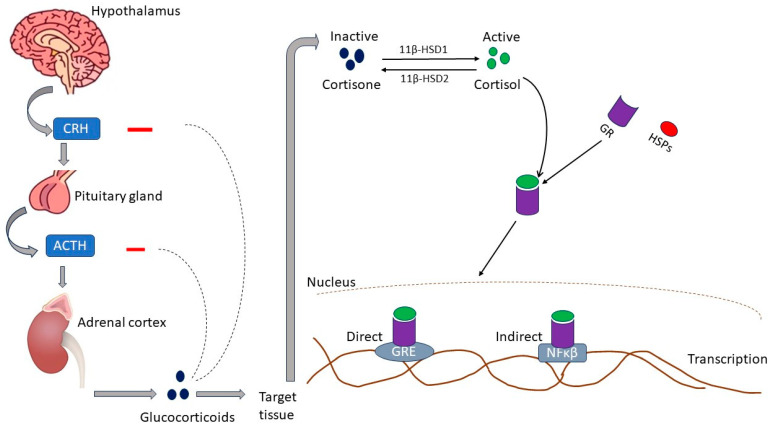
Activation of the hypothalamic–pituitary–adrenocortical axis and glucocorticoid activation and regulation. HPA axis, hypothalamic–pituitary–adrenocortical axis; CRH, corticotropin-releasing hormone; ACTH, adrenocorticotropic hormone; 11β-HSD1, 11-beta-hydroxysteroid dehydrogenase type 1; 11β-HSD2, 11-beta-hydroxysteroid dehydrogenase type 2; GR, glucocorticoid receptor; HSP, heat shock protein; GRE, glucocorticoid response element; NF-κB, nuclear factor kappa B [24].

**Figure 2 biomedicines-12-00314-f002:**
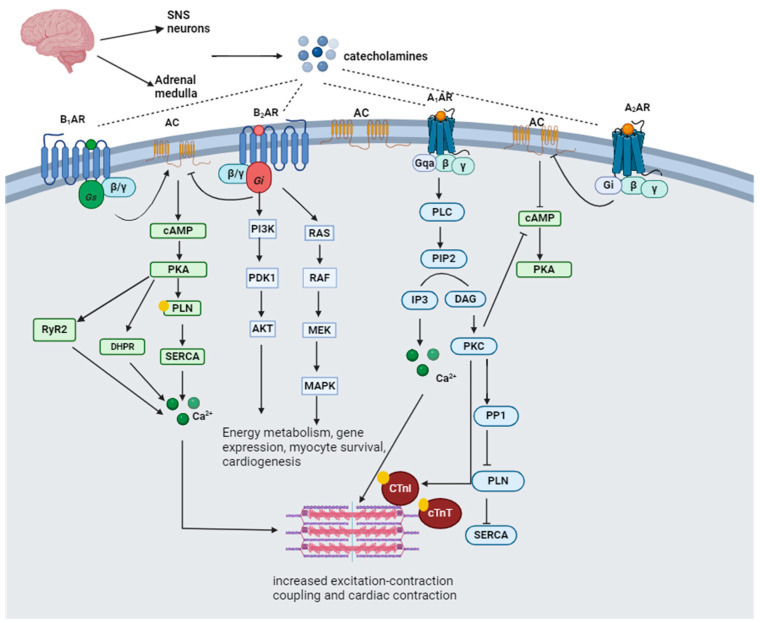
Catecholamine adrenergic signaling. Catecholamines are released from the adrenal medulla and SNS and bind to the G-protein coupled alpha- and beta-adrenergic receptors. SNS, sympathetic nervous system; B_1_AR, beta-1-adrenergic receptor; B_2_AR, beta-2-adrenergic receptor; A_1_AR, alpha-1-adrenergic receptor; A_2_AR, alpha-2-adrenergic receptor; AC, adenylyl cyclase; Gs, stimulatory G-protein subunit; Gi, inhibitory G-protein subunit; cAMP, cyclic adenosine monophosphate; PKA, protein kinase A; PLN, phospholamban; SERCA, sarcoplasmic endoplasmic reticulum calcium ATPase; RyR2, ryanodine receptor 2; DHPR, dihydropyridine receptor; Ca^2+^, calcium; PI3K, phosphoinositide-3-kinase; PDK1, phosphoinositide-dependent kinase 1; AkT, protein kinase B; RAS, Ras protein; RAF, Raf protein; MAPK, mitogen-activated protein kinase; PLC, phospholipase C; PIP2, phosphatidylinositol biphosphate; IP3, inositol triphosphate; DAG, diacylglycerol; PKC, protein kinase C; PP1, protein phosphatase 1; cTnT, cardiac troponin T; cTnI, cardiac troponin I; ↓, indicates activation; ┴ indicates inhibition; yellow circle indicates phosphorylation.

**Figure 3 biomedicines-12-00314-f003:**
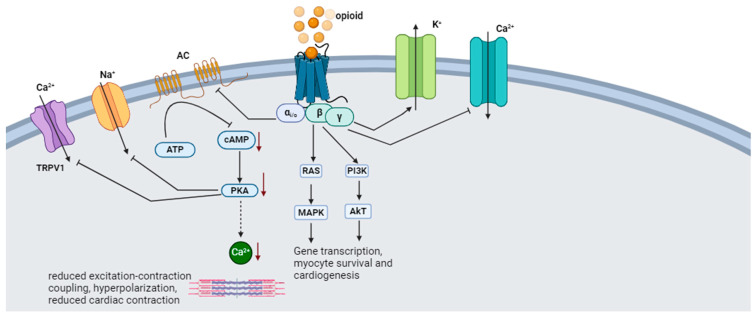
Cardiac opioid signaling. Upon release, opioids bind to G-protein-coupled opioid receptors and regulate cardiac contraction. Ca^2+^, calcium; Na^+^, sodium; K^+^, potassium; AC, adenylyl cyclase; Gα_i/o_, G protein inhibitory unit; TRPV1, transient receptor potential cation channel subfamily V member 1; cAMP, cyclic adenosine monophosphate; ATP, adenosine triphosphate; PKA, protein kinase A; RAS, Ras protein; MAPK, mitogen-activated protein kinase; PI3K, phosphoinositol-3-kinase; AkT, protein kinase B. ↓ indicates activation; ┴ indicates inhibition; ↓ (red arrow) indicates a decrease.

**Figure 4 biomedicines-12-00314-f004:**
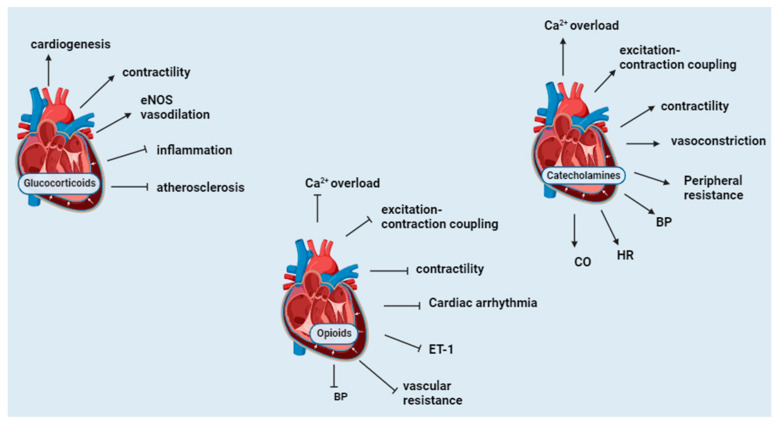
Summary of the cardiovascular function of POMC derivates. Glucocorticoids play a role in cardiogenesis, stimulate cardiac contractility and vasodilation, are anti-inflammatory, and inhibit atherosclerosis. Catecholamines cause an increase in the release of calcium, excitation–contraction coupling and contractility, peripheral resistance, and blood pressure. Opioids oppose the functions of catecholamines. eNOS, endothelial nitric oxide synthase; Ca^2+^, calcium; ET-1, endothelin 1; BP, blood pressure; HR, heart rate; CO, cardiac output. ↑ indicates an increase; ┴ indicates a decrease.

**Figure 5 biomedicines-12-00314-f005:**
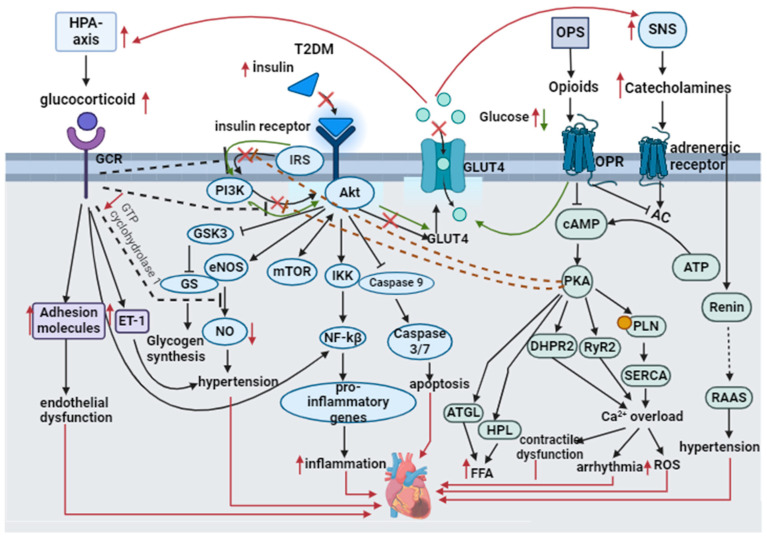
The defective PI3K/Akt pathway in T2DM results in insulin resistance and the activation of the pro-inflammatory, pro-apoptotic inhibition of eNOS and glycogen production. Insulin resistance and hyperglycemia hyperactivate the glucocorticoids and catecholamines. Glucocorticoids and catecholamines further aggravate T2DM by inhibiting the PI3K/Akt pathway, as indicated by the dotted inhibitory lines, thus potentiating insulin resistance and hyperglycemia. Glucocorticoids decrease GTP cyclohydrolase 1, which is required for NO generation from eNOS; increase ET-1 adhesion molecules; and activate NF-kβ, thus resulting in hypertension, endothelial dysfunction, and inflammation. These increase the risk of MI. Catecholamines result in Ca^2+^ overload by increasing the phosphorylation of PLN, activating SERCA calcium release, and causing leakage of RyR2 and DHPR2 receptors. Ca^2+^ overload results in mitochondrial ROS, contractile dysfunction, and arrhythmias, and these are reported to cause MI. Catecholamines also stimulate the release of renin and the activation of RAAS. Hyperactivation of the RAAS causes hypertension and increases the risk of MI. The effect of insulin resistance and hyperglycemia on opioids is not apparent, but studies have reported that opioids are elevated in hyperglycemia. The green arrows indicate that opioids activate insulin’s PI3K/Akt pathway, GLUT4 receptor translocation to the membrane, and reduced blood glucose. Opioids also inhibit adenylyl cyclase and cAMP, thus inhibiting the PKA pathway and its downstream effects. Black ↓ indicates an activation or stimulation in the pathway. Red ↓ indicates a decrease. Red ↑ indicates a increase. A black solid ┴ indicates an inhibition. X indicates defects in the insulin signaling pathway in T2DM. Black dashed lines ending with ┴ show inhibition due to the HPA-axis pathway. Brown dashed lines ending with ┴ indicate inhibition due to opioid activation. Green arrows illustrate that opioids reduce blood glucose by activating the PI3K/Akt pathway.

## Data Availability

Not applicable.

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
