# Peer review of "The Role of Pro-Opiomelanocortin Derivatives in the Development of Type 2 Diabetes-Associated Myocardial Infarction: Possible Links with Prediabetes"

_biomedicines, 2024, doi:10.3390/biomedicines12020314_

Round 1

Reviewer 1 Report

Comments and Suggestions for Authors

The authors (Gumede & Khathi) reviewed the role of POMC derivatives (as regulators of the HPA axis and its stress) in the development of a possible link during prediabetes with the occurrence of acute myocardial infarction. It seems to be accepted that obesity, then diabetes and finally myocardial dysfunction are risks of death in patients.  Too many generalizations in this review have been produced with the triad of GCs, TCs and opiates presented in parallel. The same applies to myocardial infarction (MI) and diabetes. However, the risks, causes and consequences are mixed and lack robust evidence. The role of these POMC derivatives in cardiovascular function has not been distinguished between experimental and clinical studies. Uncertainties and weight of evidence are undefined. A general signaling pathway for TCs and opiates is presented as a cardiac mechanism of action. The presentation of myocardial infarction is also patchy (apoptosis as a cell death pathway is not mentioned). A summary in Figure 4 is presented for each POMC derivative GT, CT and opioids without at least an explanatory summary. Finally, the review focuses on GC, CT and opiates in diabetes-induced myocardial infarction. Only figure 5 attempts to summarize some data on the defective pathways that could lead to myocardial infarction. As WL MIller said in his recent review, « the adrenal and pituitary glands were identified by classical anatomists, but most of this history has unfolded fairly recently and has involved complex chemical, biochemical, genetic and clinical research ». So understanding the regulation of the hypothalamic-pituitary-adrenal (HPA) axis is challenging as circadian rhythms and pulsatile GC production and their receptors to organic and cellular targets (Lightman SL, et al. Dynamics of ACTH and Cortisol Secretion and Implications for Disease). The review is a long read, with 214 references and 5 figures. This review is unconvincing and remains a speculative hypothesis without tangible evidence links with prediabetes). It should announce itself as a hypothesis and focus on the key points and present the uncertainties in order to define concrete perspectives.

Author Response

Dear Reviewer

Thank you for taking the time to review the manuscript. Please see the attachment for responses.

Thank You

Reviewer 2 Report

Comments and Suggestions for Authors

The present manuscript entitled “The Role of Proopiomelanocortin Derivatives in the Development of 2 Type 2 Diabetes-Associated Myocardial Infarction: Possible Links 3 with Prediabetes” has scientific potential for the future researcher and clinicians. Authors must rewrite the introduction to include more recent study which will give more strength for this manuscript. Authors have to focus on the clinical relevance and novelty of the current manuscript. There are some typo and grammatical errors in the text. Authors has to rewrite the conclusion which will give scientific direction to the future researcher and clinicians in the current field diabetes and cardiovascular disorders. 

Comments on the Quality of English Language

English language needs improvement.

Author Response

Dear Reviewer

Thank you for taking the time to review the manuscript. Please see the attachment below for responses.

Thank You

Reviewer 3 Report

Comments and Suggestions for Authors

The manuscript entitled  "The Role of Proopiomelanocortin Derivatives in the Development of 2 Type Diabetes-Associated Myocardial Infarction: Possible Links with Prediabetes"‚ analyse the stage for an intricate exploration into the relationship between stress, prediabetes, and the risk of myocardial infarction (MI) by elucidating the involvement of proopiomelanocortin (POMC) derivatives. The interplay between stress and diabetes-related factors, particularly the activation of the hypothalamic-pituitary-adrenocortical (HPA) axis, sympathetic nervous system (SNS), and endogenous opioid peptides (OPS), is meticulously outlined. This manuscript offers an extensive exploration of the multifaceted relationship between myocardial infarction (MI), myocardial ischemia, and various contributing factors such as atherosclerosis, inflammation, stress, and their connections with type 2 diabetes mellitus (T2DM). The document meticulously dissects the intricate pathways and molecular mechanisms implicated in the progression from myocardial ischemia to MI and the subsequent role of stress and T2DM in exacerbating cardiovascular risks.

 Strengths:

a.The manuscript provides an in-depth examination of the pathophysiological processes involved in MI, spanning from the initial stages of atherosclerosis to the impact of stress and T2DM on exacerbating cardiovascular risks.

b.It delves into intricate molecular pathways, discussing specific proteins (e.g., high mobility group box-1, NLRP3/Caspase-1/IL-1β pathway) and their roles in myocardial inflammation and fibrosis, offering a detailed understanding of the underlying mechanisms.

c.Figures accompanying the manuscript seem to provide visual aids elucidating the intricate signaling pathways involved in stress response, adrenergic signaling, and cardiac opioid signaling, enhancing the comprehension of these complex mechanisms.

I have several concerns that need to be addressed by the authors:

1.While the depth of information is commendable, the document might benefit from better organization or segmentation to improve readability and comprehension. Grouping related sections or using subheadings could aid in navigating the complex content.

2. Some sections might be challenging to grasp for readers less familiar with the subject matter. Simplifying complex pathways and terminology without compromising accuracy could enhance accessibility for a broader audience.

3. Consider using simpler language or providing brief explanations of technical terms to aid readers who might not be experts in the field.

4. Each section might benefit from a brief summary or concluding remarks that synthesize the key findings or implications discussed. This can reinforce the main points and aid in retention for the reader.

5. Ensure smooth transitions between different sections to maintain the flow of the manuscript. Clear transitions help readers follow the logical progression of ideas and connections between various aspects discussed.

These suggestions aim to improve the overall structure, clarity, and accessibility of the manuscript, making it more reader-friendly without compromising the depth of information provided.

Overall, the manuscript provides a wealth of information regarding the intricate interplay of factors contributing to MI and its exacerbation in the presence of stress and T2DM. With clearer organization and potentially simplified explanations, this work could serve as a valuable resource for researchers and clinicians seeking an in-depth understanding of these interconnected pathways.

Author Response

(The authors gave the same response as above.)

Round 2

Reviewer 1 Report

Comments and Suggestions for Authors

thank you for the attentive and clear effort for revision. The manuscript is now improved.

Reviewer 3 Report

Comments and Suggestions for Authors

I have carefully reviewed the revisions made by the authors in response to my earlier suggestions. I am pleased to confirm that the authors have diligently addressed the concerns raised, resulting in a significantly improved manuscript. The modifications include:

·       The authors have restructured the document with clear subheadings, improving the overall readability and comprehension;

·       Complex pathways and terminology have been simplified without compromising accuracy, making the manuscript more accessible to a broader audience.

·       The authors have incorporated simpler language and provided concise explanations for technical terms, ensuring broader engagement with the material.

·       Transitions between sections have been refined, maintaining a seamless flow throughout the manuscript.

Given the comprehensive and thoughtful revisions made by the authors, I recommend that the manuscript be considered for publication. The improvements contribute to a more reader-friendly document without sacrificing the depth of information.